# Silencing of the lncRNA Zeb2-NAT facilitates reprogramming of aged fibroblasts and safeguards stem cell pluripotency

Bruno Bernardes de Jesus [1], Sérgio Pires Marinho[1], Sara Barros[1], António Sousa-Franco[1], Catarina Alves-Vale[1], Tânia Carvalho[1] & Maria Carmo-Fonseca[1]

Aging imposes a barrier to somatic cell reprogramming through poorly understood mechanisms. Here, we report that fibroblasts from old mice express higher levels of Zeb2, a transcription factor that activates epithelial-to-mesenchymal transition. Synthesis of Zeb2 protein is controlled by a natural antisense transcript named Zeb2-NAT. We show that transfection of adult fibroblasts with specific LNA Gapmers induces a robust downregulation of Zeb2-NAT transcripts and Zeb2 protein and enhances the reprogramming of old fibroblasts into pluripotent cells. We further demonstrate that Zeb2-NAT expression is precociously activated by differentiation stimuli in embryonic stem (ES) cells. By knocking down Zeb2-NAT, we were able to maintain ES cells challenged with commitment signals in the ground state of pluripotency. In conclusion, our study identifies a long noncoding RNA that is overlapping and antisense to the Zeb2 locus as a target for rejuvenation strategies.

---

[1] Instituto de Medicina Molecular, Faculdade de Medicina, Universidade de Lisboa, 1649-028 Lisboa, Portugal. Correspondence and requests for materials should be addressed to B.Bd.J. (email: bruno.jesus@medicina.ulisboa.pt) or to M.C.-F. (email: carmo.fonseca@medicina.ulisboa.pt)

Somatic cell reprogramming resets aged cells back into an embryonic-like state, offering unprecedented opportunities for studying cellular rejuvenation. Cells from centenarian individuals have been successfully reprogrammed[1, 2] demonstrating the potential for aging reversibility at the cellular level. However, reprogramming efficiency declines with aging[3–6] possibly because old tissues accumulate senescent and genetically unstable cells that do not reprogram (reviewed by Mahmoudi and Brunet[7]). Indeed, it is well established that cell senescence, which can be triggered by multiple stresses such as DNA damage and replicative exhaustion associated with telomere shortening, causes a p53-dependent block to reprogramming[4, 8–10]. Improved reprogramming efficiency has been achieved by the knocking-down expression of senescence-associated proteins p53, p21$^{CIP1}$, p16$^{INK4A}$ and p14$^{ARF3, 10, 11}$, and microRNA-195[12]. However, these manipulations raise the question of whether it is safe to generate iPS cells from old donors by blocking natural defenses against cancer.

Here, we set out to investigate novel molecular pathways that constitute age-associated barriers for somatic cell reprogramming. The reprogramming of mouse fibroblasts is initiated by a mesenchymal-to-epithelial (MET) transition that is activated inside the nucleus[13]. In particular, Sox2/Oct4 suppresses the transcription of the epithelial-to-mesenchymal (EMT) transition mediator Snail, c-Myc downregulates TGF-β receptors, and Klf4 induces epithelial genes such as E-cadherin[13]. An additional transcription factor involved in this process is Zeb2, which activates EMT by repressing the expression of E-cadherin and other genes required for epithelial intercellular junctions[14, 15]. We found that fibroblasts from old mice express higher levels of Zeb2 compared to fibroblasts from young animals. As reprogramming requires suppression of pro-EMT signals, we hypothesized that Zeb2 overexpression contributes to the inefficient reprogramming of old fibroblasts. In human cells, Zeb2 expression is controlled by a natural antisense long noncoding RNA (lncRNA) named Zeb2-NAT, by a mechanism that involves retention of the first intron of

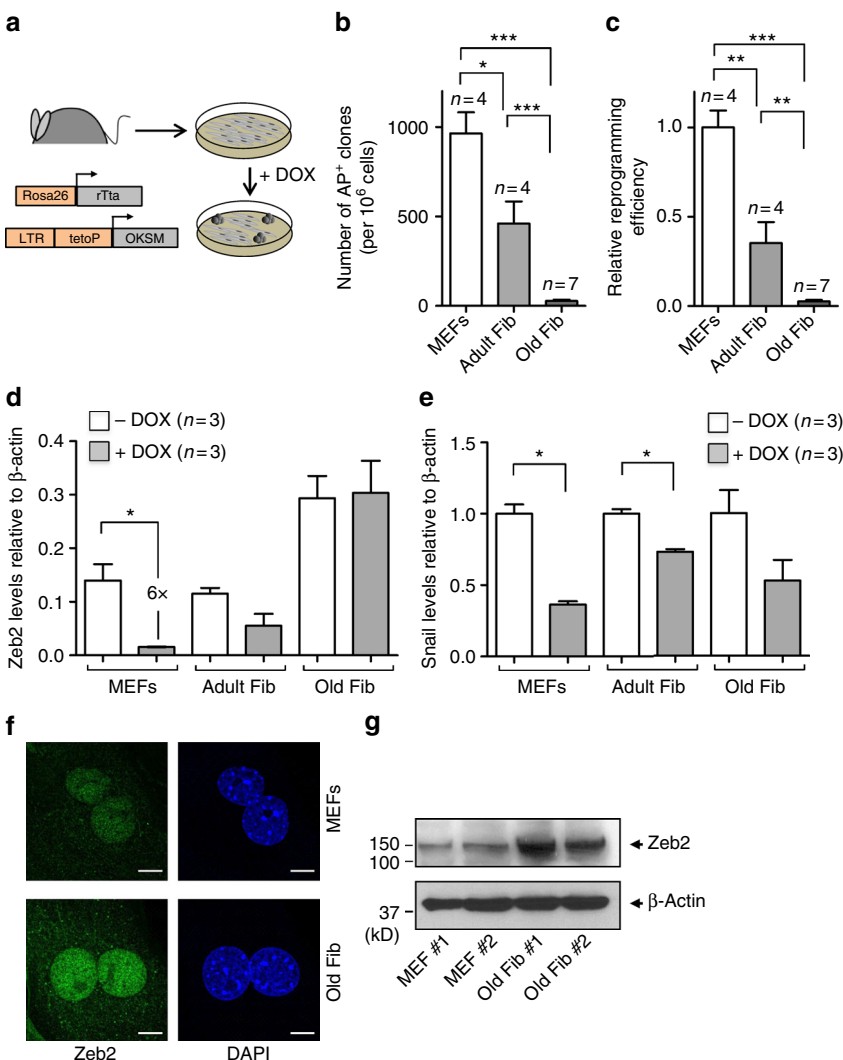

**Fig. 1** Inefficient reprograming of old fibroblasts. **a** Schematics of the reprogrammable mouse line i4F in which the OSKM reprogramming factors (Oct4, Sox2, Klf4, and c-Myc) are expressed from a single genomic locus using a doxycycline-inducible transgene. Mouse embryonic fibroblasts (MEFs) and fibroblasts isolated from adult (10–30-week old) and old (70–100-week old) mice were induced with doxycycline for 2 weeks. **b** Quantification of AP$^+$ clones. **c** Reprogramming efficiency relative to MEFs. **d, e** qRT-PCR analysis of Zeb2 (**d**) and Snail (**e**) mRNA in MEFs and fibroblasts from adult and old mice that were either not exposed to doxycycline or induced for 2 weeks. For all graphics depicted, Student's $t$ test (two-tailed) statistics, *$p < 0.05$, **$p < 0.01$, and ***$p < 0.001$; error bars represent standard deviation; $n$=experiments with independent cell cultures. **f** Immunofluorescence for Zeb2 in MEFs and old fibroblasts. The corresponding images stained with DAPI are shown (scale bars correspond to 10 μm). **g** Immunoblot for Zeb2 and β-actin in total cell lysates from two independent early-passage cultures of MEFs and old fibroblasts

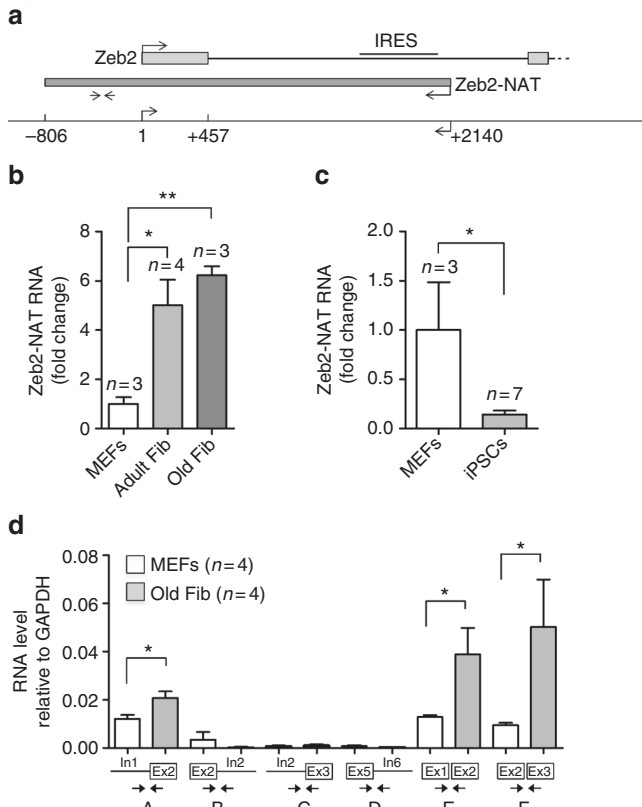

**Fig. 2** . Zeb2-NAT expression is upregulated in old fibroblasts and correlates with retention of Zeb2 intron 1. **a** Schematics of the murine *Zeb2* and *Zeb2-NAT* locus. An internal ribosome entry site (IRES) is located in the first intron of the Zeb2 transcript. Double arrows indicate the position of primers used to detect Zeb2-NAT RNA. **b** qRT-PCR analysis of Zeb2-NAT RNA in MEFs and fibroblasts from adult and old mice. Transcript levels were normalized to GAPDH mRNA and depicted as fold change relative to MEFs. **c** qRT-PCR analysis of Zeb2-NAT RNA in MEFs and in a iPS cell line derived from i4F fibroblasts. **d** qRT-PCR analysis of spliced and unspliced Zeb2 RNA in MEFs and old fibroblasts. Double arrows indicate the position of primers. For all graphics depicted, Student's *t* test (two-tailed) statistics, *$p < 0.05$, **$p < 0.01$; error bars represent standard deviation; *n*=experiments with independent cell cultures

Zeb2 pre-mRNA[16]. This intron contains an IRES sequence required for translation of Zeb2 protein, and a strong correlation was observed between expression of Zeb2-NAT, Zeb2 intron 1 retention, and Zeb2 protein synthesis[16]. Because the human and murine Zeb2 locus shares a conserved structure[17], we asked whether a similar correlation exists in mouse fibroblasts. Indeed, we observed that transfection of adult fibroblasts with specific LNA Gapmers[18] induced a robust downregulation of Zeb2-NAT transcripts and Zeb2 protein, and we demonstrate that silencing Zeb2-NAT suffices to enhance reprogramming of old fibroblasts. We also found that Zeb2-NAT expression is activated by differentiation stimuli in embryonic stem (ES) cells and we show that knocking down Zeb2-NAT maintained ES cells challenged with commitment signals in the ground state of self-renewal and pluripotency. Thus, our study unravels the potential of targeting a long noncoding antisense RNA for rejuvenation strategies.

## Results

**Zeb2 and Zeb2-NAT levels increase with aging.** We used a reprogrammable mouse line (i4F, Fig. 1a) that carries the transcriptional activator (rtTA) within the ubiquitously expressed

*Rosa26* locus and a doxycycline-inducible polycistronic cassette encoding the murine transcription factors Oct4, Sox2, Klf4, and c-Myc[19]. Upon addition of doxycycline, embryonic fibroblasts (MEFs) from i4F mice were efficiently reprogrammed in vitro based on the appearance of sharply defined cell colonies composed of small, round, and tightly packed cells that stained positive for alkaline phosphatase (AP) activity (Supplementary Fig. 1a, b). After 2 weeks of doxycycline induction, we observed an average of $936 \pm 236$ AP$^+$ clones in culture wells plated with $1 \times 10^6$ MEFs (Fig. 1b). In contrast, only $462 \pm 245$ AP$^+$ clones were observed in wells plated with the same number of early-passage (P1) fibroblasts from adult (10–30-week old) mice, and $28 \pm 21$ AP$^+$ clones were detected in wells plated with the same number of P1 fibroblasts from old (70–100-week old) animals (Fig. 1b). Quantification of reprogramming efficiency of fibroblasts isolated from adult and old mice relative to embryonic fibroblasts (MEFs) reveals a clear age-dependent decay (Fig. 1c). Accordingly, we observed that unlike fibroblasts from old mice, embryonic fibroblasts start to undergo epithelial-like morphological changes (i.e., cells became rounded, aggregated, and formed well-defined epithelial-like intercellular junctions) after 3–5 days in culture in the presence of doxycycline (Supplementary Fig. 1c). As expected, reprogramming fibroblasts showed downregulation of mRNA coding for Zeb2 (Fig. 1d) and Snail1 (Fig. 1e). However, these mRNAs were less reduced in doxycycline-treated adult and old fibroblasts compared to embryonic fibroblasts (Fig. 1d, e). The difference was particularly striking for Zeb2 mRNA, which undergoes a sixfold decrease during MEF reprogramming but is not significantly altered in adult and old fibroblasts induced with doxycycline (Fig. 1d). We further noticed that fibroblasts from aged animals express increased levels of Zeb2 protein compared to MEFs (Fig. 1f, g and Supplementary Fig. 1d). Because expression of the Zeb2 protein in human cells is controlled by the natural antisense RNA Zeb2-NAT[16], we measured the levels of Zeb2-NAT using primers that target the 3′ end of the transcript at a region without correspondence to the sequence of Zeb2 pre-mRNA (Fig. 2a), and we found that Zeb2-NAT was significantly upregulated in adult and old fibroblasts compared to embryonic fibroblasts (Fig. 2b). Moreover, reprogramming i4F fibroblasts into iPSCs resulted in significantly reduced Zeb2-NAT levels (Fig. 2c). Next, we designed primers to distinguish spliced and unspliced Zeb2 transcripts, and observed much higher levels of transcripts containing intron 1 than transcripts containing intron 2 (Fig. 2d), suggesting that splicing of the first intron of murine Zeb2 pre-mRNA is specifically regulated. Compared to MEFs, old fibroblasts have significantly higher levels of Zeb2 transcripts containing the first intron (Fig. 2d), consistent with the previously described correlation between increased expression of Zeb2-NAT and retention of the first intron in Zeb2 transcripts[16]. However, the level of Zeb2 transcripts containing either spliced intron 1 or spliced intron 2 is also higher in old fibroblasts (Fig. 2d), raising the possibility of a concomitant age-dependent upregulation of Zeb2 transcription.

**Downregulation of Zeb2-NAT enhances reprogramming of aged cells.** Having shown that Zeb2-NAT expression in fibroblasts increases with age and is reduced during reprogramming to iPSCs, we next asked whether higher levels of Zeb2-NAT transcripts contribute to the low reprogramming efficiency of aged fibroblasts. To determine the effect of reducing the levels of Zeb2-NAT on reprogramming old fibroblasts, we designed LNA Gapmers that specifically target Zeb2-NAT and Zeb2 transcripts (Supplementary Fig. 2). These are chimeric antisense oligonucleotides with a central block of DNA designed to hybridize with the target RNA. Upon hybridization, the endogenous and

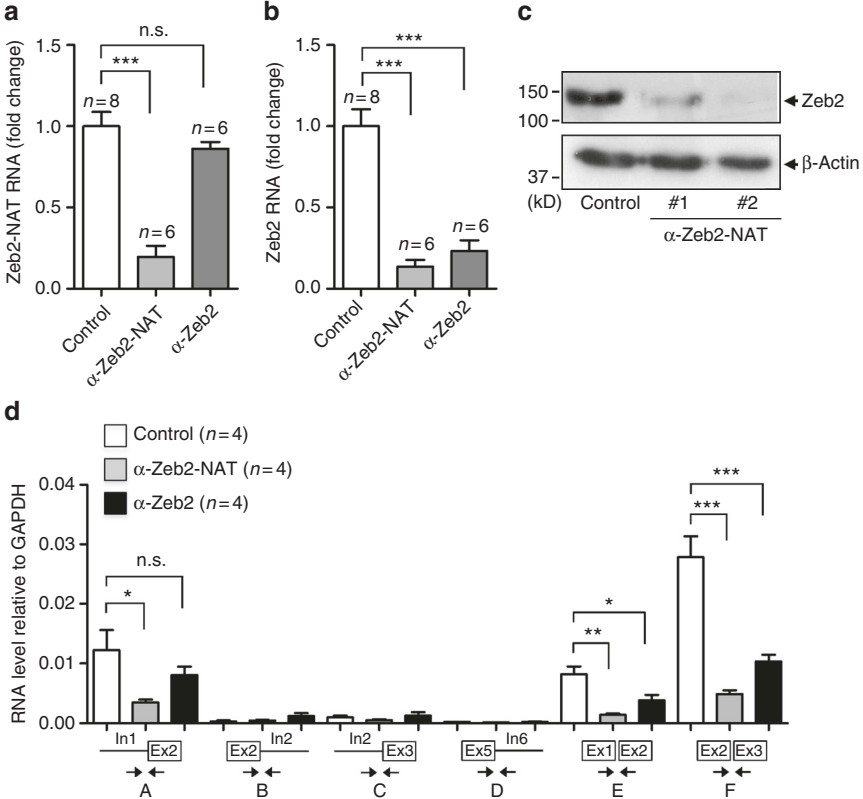

**Fig. 3** Knocking down Zeb2-NAT affects Zeb2 expression. Fibroblasts from adult mice were transfected with either control LNA Gapmers or oligonucleotides targeting Zeb2-NAT and Zeb2. **a** qRT-PCR analysis of Zeb2-NAT RNA. **b** qRT-PCR analysis of total Zeb2 transcripts (primer Zeb2 #1, see Supplementary Table 2). **c** Immunoblot for Zeb2 and β-actin in total cell lysates after transfection of anti-Zeb2-NAT Gapmers #1 and #2 (see Supplementary Fig. 2b). **d** qRT-PCR analysis of spliced and unspliced Zeb2 RNA. Double arrows indicate the position of primers. For all graphics depicted, Student's *t* test (two-tailed) statistics, *$p < 0.05$, **$p < 0.01$, ***$p < 0.001$; error bars represent standard deviation; $n$=experiments with independent cell cultures

ubiquitous ribonuclease (RNase) H hydrolyzes the RNA strand of the resulting RNA–DNA heteroduplexes. Locked nucleic acid (LNA)-modified nucleotides flanking the DNA oligonucleotide improve the stability and binding affinity for the target RNA[18]. Transfection of adult fibroblasts with antisense oligonucleotides targeting Zeb2-NAT resulted in a robust decrease of Zeb2-NAT RNA (Fig. 3a), total Zeb2 RNA (Fig. 3b), and Zeb2 protein (Fig. 3c). Transfection with antisense oligonucleotides targeting Zeb2 reduced the levels of Zeb2 transcripts (Fig. 3b) but not Zeb2-NAT (Fig. 3a). Further analysis of spliced and unspliced Zeb2 transcripts revealed that oligonucleotides targeted to Zeb2 RNA efficiently reduced the level of Zeb2 mRNA devoid of intron 1 and 2 (Fig. 3d), but had a marginal effect on the level of Zeb2 transcripts with intron 1 retention (Fig. 3d). In contrast, oligonucleotides targeted to Zeb2-NAT caused a significant reduction of both spliced Zeb2 mRNA and Zeb2 transcripts with intron 1 retention (Fig. 3d). Similar results were observed using digital droplet PCR (Supplementary Fig. 3a), which reduces PCR bias and provides a higher degree of sensitivity and precision[20, 21]. It is also evident from the results obtained by digital PCR that Zeb2-NAT RNA molecules are much less abundant than Zeb2 transcripts. To determine whether knocking down Zeb2-NAT affects Zeb2 transcription, we used metabolic labeling with modified uridine (4-thiouridine, 4sU), which allows to distinguish recently transcribed RNA from steady-state RNA levels[22–24]. Analysis of RNA-4sU shows that targeting Zeb2-NAT leads to increased levels of newly transcribed total Zeb2 RNA and lower levels of newly transcribed Zeb2 carrying the first intron (Supplementary Fig. 3b). Taken together, these experiments suggest that

knockdown of Zeb2-NAT enhances both the transcriptional rate of the *Zeb2* gene and retention of intron 1 in Zeb2 transcripts. We next transfected fibroblasts from aged i4F mice with LNA Gapmers targeting either Zeb2 or Zeb2-NAT, and 24 h later, doxycycline was added to the culture to induce reprogramming. Upon 3 weeks of culture in a medium supplemented with doxycycline, we assessed reprogramming efficiency based on the number of AP+ colonies. As shown in Fig. 4 (Supplementary Fig. 4a, b), significantly more colonies were observed in cultures of fibroblasts from adult and old mice transfected with anti-Zeb2-NAT and anti-Zeb2 oligonucleotides compared to cultures treated with nonspecific oligonucleotides. To determine whether these AP+ colonies correspond to bona fide iPS cell lines, individual colonies with a morphology similar to those formed by ES cells were picked (Supplementary Fig. 4c), and cells were transferred to a chemically defined medium containing the cytokine leukemia inhibitory factor LIF[25, 26] and the 2i small-molecule inhibitors CHIR99021 and PD0325901[27], without doxycycline and in the absence of feeder cells. Of all colonies derived from old animals, only those that had been transfected with anti-Zeb2-NAT oligonucleotides survived (Fig. 4c). The surviving colonies expressed the markers of pluripotency Nanog and SSEA1, as shown by immunofluorescence (Fig. 4d). Moreover, when cells from these colonies were subcutaneously injected into immune-deficient mice, they induced the formation of tumoral masses that are reminiscent of teratomas[28] with disorganized differentiation into endodermal, mesodermal, and ectodermal derivatives (Fig. 4e), thus confirming their in vivo differentiation potential. We therefore conclude that downregulation of Zeb2-NAT is

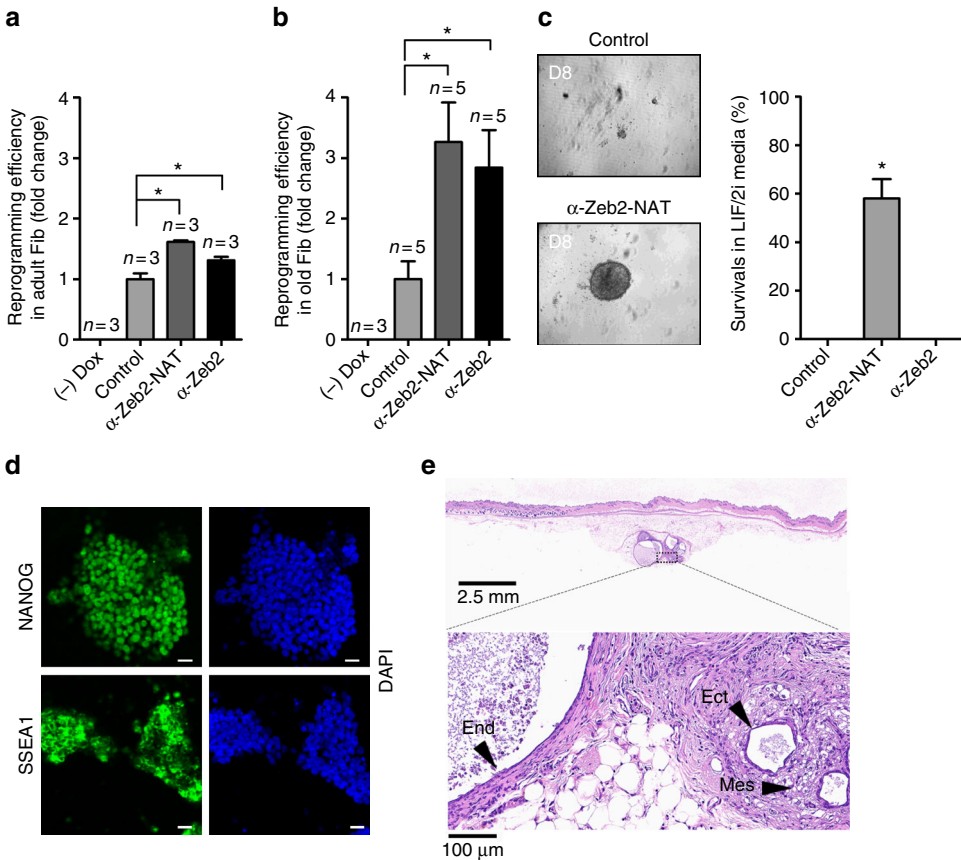

Fig. 4 Knocking down Zeb2-NAT is sufficient to enhance reprogramming of old fibroblasts. Fibroblasts were transfected with the indicated LNA Gapmers and induced to reprogram for 3 weeks. **a**, **b** Reprogramming efficiency in adult and old fibroblasts based on quantification of AP$^+$ colonies. Student's $t$ test (two-tailed) statistics, *$p < 0.05$; error bars represent standard deviation; $n$=experiments with independent cell cultures. **c** Proportion of colonies that survived re-plating in medium containing LIF and 2i, without doxycycline; colonies were counted at day 8 after replating. At least ten independent colonies where picked from each condition. Fischer's exact test was used for statistical analysis. **d** Immunofluorescence for Nanog and SSEA1 in reprogrammed old fibroblasts transfected with anti-Zeb2-NAT oligonucleotides; the corresponding images stained with DAPI are shown (scale bars, 20 μm). **e** Histological section of a subcutaneous teratoma containing mesoderm (mes), endoderm (end), and ectoderm (ect) 15 weeks after injection of cells into immune-deficient mice. Magnifications as indicated

sufficient to liberate aged fibroblasts from endogenous constraints to reprogramming without compromise to developmental potency.

Although AP$^+$ colonies derived from control old fibroblasts did not survive in feeder-free culture conditions in the absence of doxycycline (Fig. 4c), a few clones were able to grow when cultured in the presence of feeder cells. These clones expressed the markers of pluripotency Nanog and SSEA1 (Supplementary Fig. 4d) and induced the formation of teratomas, confirming that they correspond to iPSCs. These results show that knocking down Zeb2 expression enhances the reprogramming potential of old fibroblasts but is not essential for this process, which is in agreement with previous studies reporting the successful reprograming of cells from centenarian individuals[1, 2]. As an alternative to antisense Gapmer oligonucleotides, we tested the ability of short hairpin RNAs (shRNAs) to downregulate Zeb2 and Zeb2-NAT RNA by RNA interference (RNAi). Infection of fibroblasts from old mice with lentiviral particles containing a DNA sequence that codes for a shRNA targeting Zeb2 had no significant effect on Zeb2-NAT RNA (Supplementary Fig. 5a), but resulted in a robust decrease of total Zeb2 RNA (Supplementary Fig. 5b), and Zeb2 protein (Supplementary Fig. 5c). Although expression of a shRNA targeting Zeb2-NAT also decreased the levels of Zeb2 protein (Supplementary Fig. 5c), reduction of Zeb2-NAT and total Zeb2 transcripts (Supplementary Fig. 5a, b) was

much less striking than the effect observed with anti-Zeb2-NAT Gapmers (Fig. 3a, b). Possibly, the low abundant Zeb2-NAT transcripts remain mostly restricted to the nucleus and therefore are not efficiently degraded by cytoplasmic RNAi, while RNase-H-mediated Gapmer knockdown efficiently suppresses RNA targets localized both in the nucleus and in the cytoplasm[29]. As expected from the lower levels of Zeb2 protein induced by shRNAs, RNAi-mediated knockdown of Zeb2 and Zeb2-NAT expression resulted in increased reprogramming efficiency of old fibroblasts (Supplementary Fig. 5d–f). To determine whether the knockdown of Zeb2 and Zeb2-NAT transcripts affects the expression of previously known modulators of the reprogramming process[3, 8, 30], we analyzed the tumor suppressors p16, p21, and p53. Fibroblasts from old mice were transfected with control LNA Gapmers and oligonucleotides targeting Zeb2-NAT and Zeb2. The results show no significant change in either p16, p21, and p53 RNA levels (Supplementary Fig. 6a) or proliferation capacity (Supplementary Fig. 6b), indicating that Zeb2 and Zeb2-NAT are implicated in the reprogramming process through a distinct mechanism.

It was recently reported that ectopic expression of Zeb2-NAT in human cells prevents splicing of intron 1 in Zeb2 mRNA and increases Zeb2 protein synthesis[16], arguing for a direct regulatory role of the antisense RNA product rather than the act of antisense transcription. It was further proposed that the antisense RNA

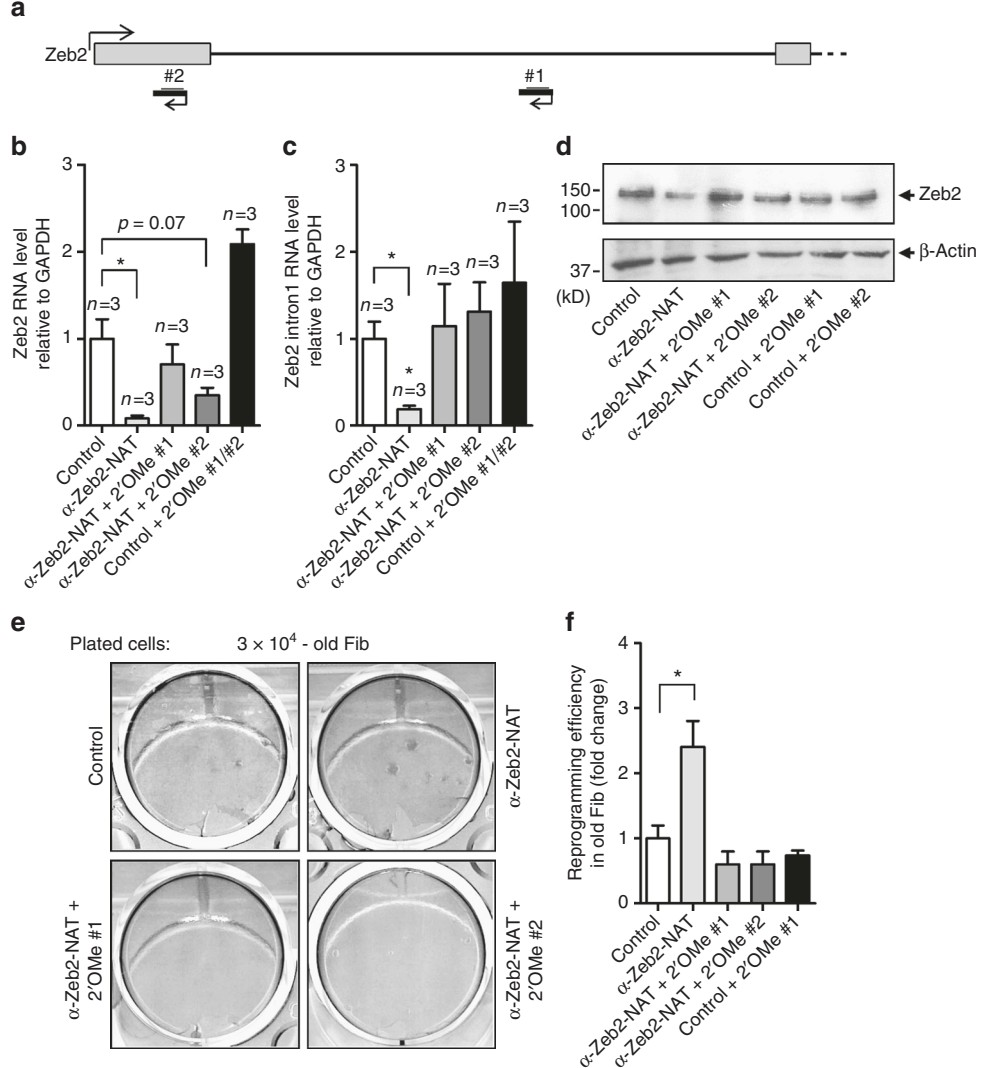

**Fig. 5** Expression of 2′OMe RNA oligonucleotides mimicking Zeb2-NAT. **a** Position of 2′OMe RNA oligonucleotides relative to Zeb2 RNA. **b**, **c** qRT-PCR analysis of Zeb2 transcripts (primers Zeb2 #1 and Zeb2 intron, see Supplementary Table 2) in fibroblasts from old mice transfected as indicated. **d** Immunoblot for Zeb2 and β-actin in total cell lysates of old fibroblasts transfected as indicated. **e** Representative reprogramming experiment stained for Alkaline Phosphatase after 3 weeks in culture with doxycycline. **f** Reprogramming efficiency of old fibroblasts transfected as indicated. For all graphics depicted, Student's $t$ test (two-tailed) statistics, *$p < 0.05$. At least three independent experiments were carried per condition

interferes with spliceosome assembly through base-pairing with sequences in Zeb2 pre-mRNA that are required for intron 1 splicing[16]. To investigate whether such a mechanism would explain the phenotype observed in murine fibroblasts, we designed two Zeb2-NAT mimics composed of 2′-O-methyl (2′OMe) RNA oligonucleotides[31] (Fig. 5a). The 2′OMe RNA oligonucleotides are complementary to Zeb2 pre-mRNA sequences predicted to be involved in splicing[32]. Upon simultaneous transfection of old fibroblasts with the LNA Gapmer anti-Zeb2-NAT and either 2′OMe RNA #1 or 2′OMe RNA #2, the levels of total Zeb2 RNA and Zeb2 transcripts with intron 1 retention were similar to those observed in untreated cells, and significantly higher than in cells transfected with LNA Gapmer anti-Zeb2-NAT (Fig. 5b, c). We further observed that after transfection with anti-Zeb2-NAT Gapmer and 2′OMe RNAs, the level of Zeb2 protein was similar to control and higher than in cells transfected with Gapmer only (Fig. 5d). Transfection of old fibroblasts with 2′OMe-RNAs in the absence of anti-Zeb2-NAT Gapmer did not cause statistically significant changes, although the levels of Zeb2 transcripts tended to be increased relative to

control, untreated cells (Fig. 5b, c). Taken together, these results indicate that 2′OMe RNAs can rescue the knockdown of endogenous Zeb2-NAT transcripts. We therefore reasoned that if 2′OMe RNAs act as functional mimics of Zeb2-NAT, they should prevent the increase in reprogramming efficiency caused by Zeb2-NAT downregulation. Indeed, the reprogramming efficiency was similar in control, untreated old fibroblasts and in cells simultaneously transfected with the LNA Gapmer anti-Zeb2-NAT and either 2′OMe-RNA #1 or 2′OMe RNA #2, in contrast with the higher efficiency observed in old fibroblasts transfected with the LNA Gapmer only (Fig. 5e, f).

**Zeb2-NAT expression correlates with stem cells commitment**. Having shown that downregulation of Zeb2-NAT enhances reprogramming of old fibroblasts into iPSCs, we next asked whether this antisense transcript also plays a role in stem cell pluripotency. First, we analyzed the expression of Zeb2-NAT in the murine ES cell line E14Tg2a.4 (E14)[33] and found that Zeb2-NAT transcripts are expressed at very low levels (Fig. 6a),

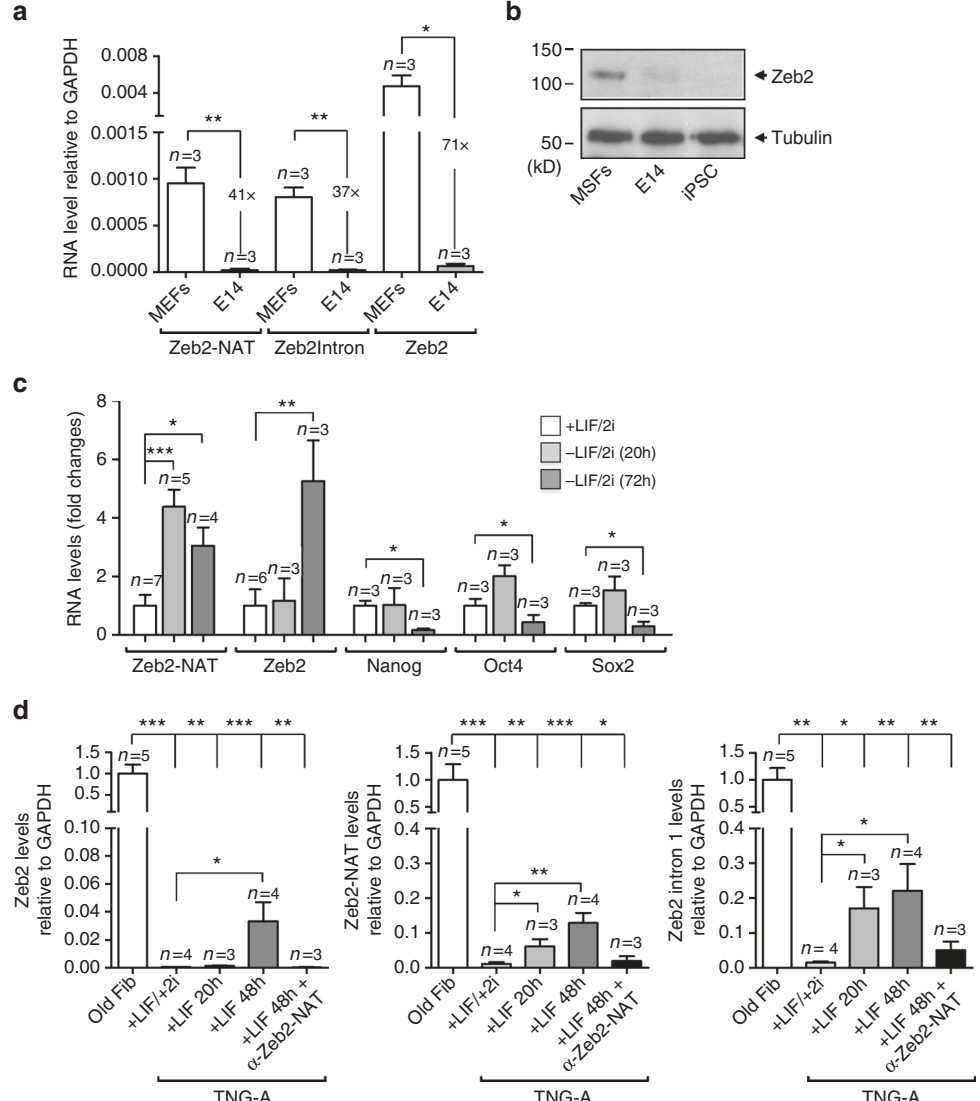

**Fig. 6** Zeb2-NAT expression in ES cells increases in response to differentiation stimuli. **a** qRT-PCR analysis of Zeb2-NAT and Zeb2 transcripts in MEFs and in the embryonic stem cell line E14. **b** Immunoblot for Zeb2 and tubulin in total cell lysates from MSFs (mouse skin fibroblasts), E14 cells and iPSCs. **c** qRT-PCR analysis of the indicated transcripts in E14 cells cultured in medium with LIF/2i or incubated for 20 and 72 h after removal of LIF and 2i. **d** qRT-PCR analysis of the indicated transcripts in old fibroblasts and TNG-A cells grown in the presence or absence of 2i. For all graphics depicted, Student's $t$ test (two-tailed) statistics was used, $*p < 0.05$, $**p < 0.01$, $***p < 0.001$; error bars represent standard deviation; $n$=experiments with independent cell cultures

similarly to that observed in iPSCs derived from i4F fibroblasts (Fig. 2c). Compared to MEFs, the level of Zeb2-NAT is 41-fold lower in ES cells (Fig. 6a). The level of total Zeb2 RNA and Zeb2 transcripts with retention of the first intron is also significantly lower in ES cells (Fig. 6a), and Zeb2 protein is not detected by immunoblot (Fig. 6b). The observation that Zeb2-NAT levels are much lower in pluripotent cells compared to fibroblasts suggests that increased expression of this antisense RNA may be required during the differentiation process. To determine whether Zeb2-NAT expression increases when stem cells shift from self-renewal to lineage commitment, E14 cells were challenged by removing LIF and 2i from the culture medium. The combined presence of these inhibitors, which act by shielding cells from inductive differentiation stimuli, is required to maintain ES cells in the ground state of self-renewal and pluripotency[27]. We found that at 20 h after LIF/2i removal, Zeb2-NAT was already significantly upregulated (Fig. 6c). At the same time point, the levels of Zeb2, Nanog, Oct4, and Sox2 transcripts remained unchanged.

Downregulation of the pluripotency genes Nanog, Oct4, and Sox2 was detected later, after 72 h of culture in a medium without LIF/ 2i (Fig. 6c). At the 72-h time point, the level of total Zeb2 transcripts increased (Fig. 6c), as expected since Zeb2 expression is higher in differentiated cells than in pluripotent cells. Additionally, we took advantage of the fact that ES cells grown to high confluence tend to spontaneously differentiate. Consistent with the view that loss of pluripotency correlates with activation of Zeb2-NAT expression, cells grown to high confluence expressed higher levels of Zeb2-NAT transcripts compared to cells grown at low density (Supplementary Fig. 7a, b). In contrast, the level of total Zeb2 RNA remained similar in cells grown at low and high confluence (Supplementary Fig. 7c), providing further evidence that activation of Zeb2-NAT expression precedes upregulation of Zeb2 expression. These results argue that Zeb2-NAT transcription in ES cells is tightly controlled by differentiation stimuli and that activation of Zeb2-NAT expression represents an early marker for loss of pluripotency.

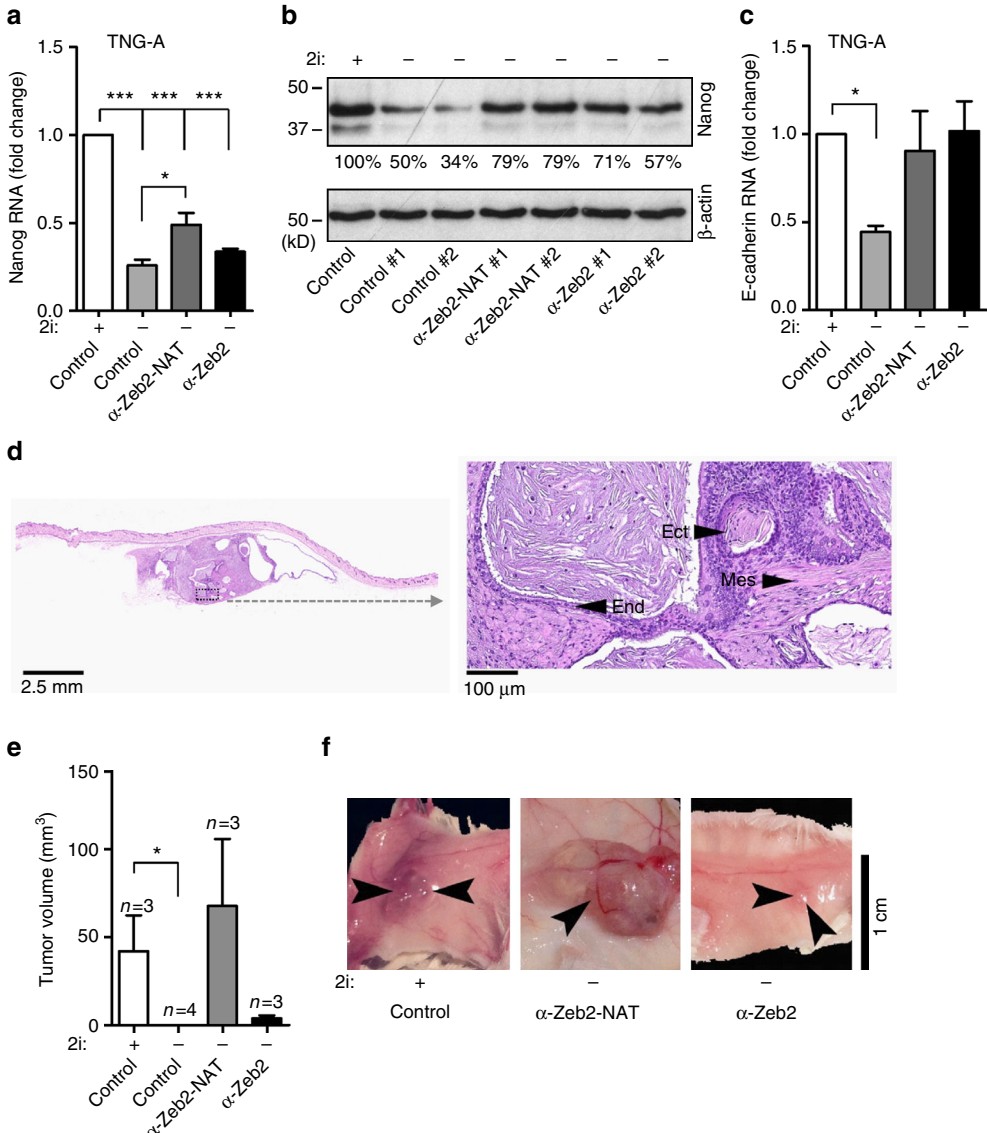

**Fig. 7** Blocking expression of Zeb2-NAT in ES cells enhances pluripotency and self-renewal. **a** qRT-PCR analysis of Nanog RNA in TNG-A cells transfected with the indicated oligonucleotides and grown in the presence (+) or absence (−) of 2i for 48 h. Transcript levels were normalized to GAPDH mRNA and depicted as fold change. Student's $t$ test (two-tailed) statistics, $*p < 0.05$, $***p < 0.001$; error bars represent standard deviation. At least three independent experiments were carried per condition. **b** Immunoblot Nanog and β-actin in total cell lysates from TNG-A cells transfected as indicated and grown in the presence (+) or absence (-) of 2i for 48 h. Relative quantification was carried out using Scion Image Software. **c** qRT-PCR analysis of E-cadherin RNA in TNG-A cells transfected as indicated and grown in the presence (+) or absence (−) of 2i for 48 h. Transcript levels were normalized to GAPDH mRNA and depicted as fold change. Student's $t$ test (two-tailed) statistics, $*p < 0.05$; error bars represent standard deviation. At least three independent experiments were carried per condition. **d** Histological section of a subcutaneous teratoma containing mesoderm (mes), endoderm (end), and ectoderm (ect), at 7 weeks after injection of cells treated with anti-Zeb2-NAT oligonucleotides. **e** Quantification of tumor volume (mm³). Student's $t$ test (two-tailed) statistics, $*p < 0.05$; error bars represent standard deviation; $n$=teratomas from independent cell cultures. **f** Representative images of tumors observed at 7 weeks after cell transplantation. Magnifications as indicated

**Blocking Zeb2-NAT safeguards ES cells pluripotency**. We additionally investigated how activation of Zeb2-NAT expression relates to loss of pluripotency in ES cells challenged by removal of 2i from the culture medium. We analyzed E14 cells and the transgenic ES strain TNG-A that carries a green fluorescence protein (GFP) reporter inserted into one Nanog allele (Nanog $^{+/GFP}$)[34]. It was previously reported that TNG-A cells are more prone to exit the ground state of pluripotency[35]. Accordingly, and in contrast to E14 cells, TNG-A cells maintained in a medium without 2i rapidly changed their morphology (Supplementary Fig. 8a, b). We observed that 20 h after removing 2i from the culture medium of TNG-A cells, Zeb2-NAT expression was

already significantly upregulated (Fig. 6d). The level of Zeb2 transcripts retaining intron 1 was also increased at 20 h in culture without 2i (Fig. 6d), consistent with the proposed role of Zeb2-NAT in preventing splicing of the first Zeb2 intron. The total level of Zeb2 RNA was similar to control at 20 h, but increased at 48 h (Fig. 6d), in further agreement with the view that upregulation of Zeb2-NAT expression precedes activation of Zeb2 transcription. Transfection with LNA Gapmers targeting Zeb2-NAT blocked upregulation of Zeb2 expression, in agreement with the proposed model that ZEB2-NAT expression activates Zeb2 transcription. Both E14 and TNG-A cells grown in a medium supplemented with 2i presented a robust expression of Nanog

RNA (Fig. 7a) and protein (Fig. 7b and Supplementary Fig. 8c). At 48 h after removal of 2i from the culture medium, the levels of Nanog decreased, particularly in TNG-A cells (Fig. 7a, b and Supplementary Fig. 8c). The levels of E-cadherin were also significantly decreased in TNG-A cells cultured without 2i (Fig. 7c). Transfection with anti-Zeb2-NAT or anti-Zeb2 oligonucleotides was sufficient to counteract this effect (Fig. 7a–c). Namely, in TNG-A cells maintained for 48 h without 2i but transfected with anti-Zeb2-NAT or anti-Zeb2 oligonucleotides, the levels of E-cadherin mRNA were similar to those detected in cells cultured in a medium with 2i (Fig. 7c). Compared to E14 cells (Supplementary Fig. 8d), TNG-A cells grown in a medium without 2i additionally showed reduced capacity to form teratomas upon injection into immune-deficient mice (Fig. 7d–f and Supplementary Fig. 9). Although tumors were detected in animals injected with cells pretreated with both anti-Zeb2-NAT and anti-Zeb2 oligonucleotides, larger and more differentiated teratomas were observed when Zeb2-NAT was targeted (Fig. 7d–f and Supplementary Fig. 9). This could be because LNA Gapmers were more efficient in knocking down Zeb2-NAT transcripts than Zeb2 mRNA. Altogether, these results suggest that blocking Zeb2-NAT upregulation in stem cells was challenged with differentiation stimuli safeguards pluripotency. E14 cells are probably better shielded against differentiation stimuli than TNG-A cells, and thus maintain the ground state of self-renewal and pluripotency when cultured for 48 h in the absence of 2i. However, a decrease in expression of the pluripotency marker Oct4 was observed in E14 cells maintained in a medium without 2i at a later time point (96 h) (Supplementary Fig. 10). When E14 cells were first transfected with anti-Zeb2-NAT Gapmer (Supplementary Fig. 10a, b) and then cultured for 96 h in a medium without 2i, the level of Oct4 was significantly higher (Supplementary Fig. 10c). We also observed that transfection of E14 cells with anti-Zeb2-NAT and anti-Zeb2 Gapmers before culture for 96 h in the absence of 2i yielded more AP+ cells (Supplementary Fig. 10d). This suggests that, as observed in TNG-A cells, blocking Zeb2-NAT expression in E14 cells challenged by removal of 2i from the culture medium maintains self-renewal and pluripotency.

## Discussion

Here, we describe antisense transcription as a novel regulatory layer with implications in reprogramming efficiency of aged cells and conservation of pluripotent features. Sense and antisense transcripts of the same locus have been shown to be common occurrences[36]. Yet, few antisense lncRNAs have been shown to be functional[37, 38]. Our results show that a lncRNA that is overlapping and antisense to the Zeb2 locus regulates expression of the Zeb2 protein in mouse fibroblasts and ES cells. Reducing the levels of Zeb2-NAT RNA facilitates OSKM-induced iPSC generation of aged cells by liberating fibroblasts from endogenous constraints without compromising their developmental potency. Additionally, we found that Zeb2-NAT expression is rapidly upregulated when ES cells receive commitment signals, and blocking the expression of this antisense transcript is sufficient to maintain challenged ES cells in a state of self-renewal and pluripotency. In summary, our results underscore the physiological role of a long noncoding antisense RNA in fine-tuning the expression of a protein-coding gene implicated in pluripotency, differentiation, and reprogramming.

## Methods

**Fibroblast reprogramming**. Fibroblasts were derived from the reprogrammable transgenic mouse line i4F[19]. MEFs were prepared from total mouse embryos as previously described[39]. Adult (10–30-weeks old) and old (70–100-weeks old) fibroblasts were obtained from the mouse ears as described[40]. For reprogramming, passage 1 fibroblasts were plated at a density of $5 \times 10^5$ cells per well in six-well gelatin-coated plates, or as stated in the figure legends, and cultured in iPS cell medium with doxycycline (1 µg ml$^{-1}$). The medium was changed every 48 h until iPS-like cell colonies appeared (after ~7 days of treatment). Reprogramming plates were stained for AP activity (AP detection kit, Chemicon International).

**Culture conditions**. Cells were maintained in an incubator at 37 °C and 5% CO$_2$. Primary fibroblasts were cultured in Dulbecco's modified Eagle's medium (DMEM; Gibco) containing 10% fetal bovine serum (FBS; Promocell), 2 mM GlutaMAX (Gibco), and 100 µg/ml penicillin–streptomycin (Sigma). E14, TNG-A, and iPS cells were maintained in feeder-free conditions using gelatin-coated plates (0.1% gelatin) and iPS cell medium that contains high-glucose DMEM supplemented with KSR (15%, Invitrogen), LIF (1000 U/ml), nonessential amino acids, penicillin–streptomycin, glutamax, and β-mercaptoethanol. The 2i chemical inhibitors CHIR99021 and PD0325901 were purchased from Axon Medchem, dissolved in DMSO, and used at 3 µM and 250 µM, respectively.

**LNA Gapmers**. RNase H-activating Gapmers were designed and purchased from Exiqon. Antisense oligonucleotides with a phosphorothioate backbone and sequences complementary to *Mus musculus* Zeb2-NAT (Genbank accession no. NR_110572.1) and Zeb2 (Genbank accession no. NM_001289521.1) were flanked by LNA-modified bases. As negative control, we used the antisense LNA Gapmer provided by Exiqon (172392—seq: AACACGTCTATACGC), which is designed to be similar to specific Gapmers but has no homology to any known microRNA, lncRNA, or mRNA sequence in the mouse. The sequences are detailed in Supplementary Table 1. LNA Gapmers were transfected twice with a 24-h interval using Lipofectamine RNAiMAX (Invitrogen) and used at a final concentration of 10 mM per transfection. Unless otherwise stated in figure legends, cells were analyzed one day after the last transfection.

**shRNAs**. From Sigma Mission, we purchased sequence-verified viral vector against mZeb2 (TRCN0000070887 and TRCN0000070883) and was custom made against mZeb2-NAT (the sequences are detailed in Supplementary Table 1) in the pLK0.1-puro-CMV-tGFP vector. Lentiviral particles against the same transcript were mixed equimolarly. For cell transduction, we used lentiviral particles at a multiplicity of infection of 5, overnight, as previously described[41]. Briefly, cells were plated at a final confluency of 50–60% in a six-well plate the day before infection. The virus was mixed in culture media with 8 µg/ml of polybrene and added directly to the cells. After incubation for 24 h, the media was changed and cells were harvested 48–72 h post infection.

**2′OMe-RNA oligonucleotides**. 2′-OMe RNA oligonucleotides mimicking different regions of Zeb2-NAT were synthesized by Sigma. The sequences of the oligonucleotides are shown in Supplementary Table 1. 2′-OMe RNA oligonucleotides were transfected twice with a 24-h interval using Lipofectamine RNAiMAX (Invitrogen), at a final concentration of 10 mM per transfection.

**4sU pulse**. 4sU was added to the cell culture medium for 30 min to a final concentration of 500 µM. Afterward, the medium was removed and the cells were immediately lysed with Trizol. Newly transcribed tagged RNA fractions were isolated as previously described[23, 42]. Briefly, RNA was labeled with Biotin-HPDP (2 µl of Biotin-HPDP (Pierce) (1 mg/ml dimethylformamide) per 1 µg of RNA) for 1.5 h at room temperature and further precipitated with chloroform/isoamyl alcohol (24:1). To separate labeled and unlabeled RNA, we used streptavidin-coated magnetic beads. Biotinylated RNA samples were heated to 65 °C for 10 min and immediately placed on ice for 5 min. Biotinylated RNA was mixed with streptavidin beads (1:1 v/v) and incubated at RT for 10 min with rotation. RNA was washed (wash buffer 1×: 100 mM Tris, pH 7.5; 10 mM EDTA; 1 M NaCl; and 0.1% Tween20) into µMacs columns in a magnetic strand and eluted (100 µl of elution buffer (100 mM DTT)) directly into RLT buffer. RNA was further recovered with a RNAeasy MinElute Spin Column before analysis.

**RNA analysis**. Total RNA was extracted from cells with Trizol (Life Technologies), DNase I treated, and retrotranscribed into cDNA using random hexamers as primers following the manufacturer's protocol (Roche First-Strand cDNA Synthesis Kit). Quantitative real-time PCR was performed using Syber Green Power PCR Master Mix (Applied Biosystems) in an Applied Biosystems 7500 fast or ViiA 7 equipment. For input normalization, we used the housekeeping genes Actb (β-actin) and Gapdh. The primers used are listed in Supplementary Table 2. The relative expression of each RNA of interest was determined by calculating $2^{-\Delta\Delta ct}$ values, which express the difference between the cycle threshold for the primer pair targeting the RNA of interest and the primer pair targeting a housekeeping gene. Quantitative PCR data were obtained from independent biological replicates ($n$ values indicated in the corresponding figures) and were tested for normal distribution using the Shapiro–Wilk test and for equal variance using the $F$ test. Normal distribution and equal variance were confirmed in the large majority of data and, therefore, we assumed normality and equal variance for all samples. Based on this, we used the Student's $t$ test (two-tailed, unpaired) to estimate statistical significance.

**Droplet digital PCR**. Droplet digital PCR (ddPCR) was carried out according to manufacturer recommendations and as described[43, 44]. Briefly, the ddPCR mixture contained 7.5 µl of 2× ddPCR Evagreen Supermix (Bio-Rad, Hercules, USA), 10 nM of both the Zeb2-NAT, Zeb2 intron 1, or GAPDH forward and reverse primers, and 5 ng of cDNA (RNA equivalent) in each 15-µl reaction. The 15-µl reaction was assembled into a droplet cartridge (Bio-Rad, Hercules, USA), according to the Bio-Rad protocol and the cartridge was placed in the droplet generator (Bio-Rad #186-3002) giving rise to around 20,000 individual droplets in an emulsion[43]. Droplets were transferred to a 96-well plate (Eppendorf, Hamburg, Germany) and thermal sealed with a foil lid (Eppendorf, Hamburg, Germany). Sealed plates were cycled using a Veriti DX thermal cycler (ThermoFisher Scientific) under the following conditions: 2 min at 30 °C; 10-min hold at 95 °C; 48 cycles of 95 °C for 50 s and then 59 °C for 120 s; 5 min at 4 °C; and 5 min at 90 °C. After amplification, the plate was transferred to a Bio-Rad droplet reader (QX200) from which data were extracted and analyzed with the Quantasoft software following manufacturer recommendations.

**Protein analysis**. Whole-cell extracts were prepared in lysis buffer (150 mM NaCl, 50 mM Tris, pH 8, 1% NP40, 0.5% sodium deoxycholate, 0.1% SDS, and protein inhibitors cocktail). A total protein amount of 30 µg was loaded per lane in a PAGE 10% Bis–Tris gel of 1.0 mm and electrophoresed in MES SDS Running Buffer (Invitrogen). The antibodies used for immunoblotting are listed in Supplementary Table 3 and the full blots are depicted in Supplementary Fig. 11.

**Immunofluorescence**. Cells previously seeded on coverslips were fixed with 2% paraformaldehyde for 15 min at room temperature, washed with PBS, and permeabilized (0.02% Triton X-100 in PBS) for 20 min. Samples were then blocked for 1.5 h in PBS containing 5% BSA, incubated for 2 h with primary antibodies diluted in PBS containing 4% BSA, washed with PBS containing 0.01% Tween20, and incubated for 2 h with secondary antibodies conjugated with Cy3 or Alexa488 (diluted in PBS with 4% BSA). The primary antibodies used for immunofluorescence are listed in Supplementary Table 3. Cells were imaged using Zeiss confocal microscope (Zeiss LSM 880).

**Animal procedures**. Animal experimentation at Instituto de Medicina Molecular was conducted strictly within the rules of the Portuguese official veterinary directorate, which complies with the European Guideline 86/609/EC concerning laboratory animal welfare, according to a protocol approved by the institute's Animal Ethics Committee. To assess the capacity for teratoma formation, cells were trypsinized and $2 \times 10^6$ cells were subcutaneously injected into the flanks of 8-week-old immunocompromised mice (NOD-SCID mice from Charles River). Animals were killed with anesthetic overdose and necropsy was performed. Subcutaneous tumor and ipsilateral inguinal lymph node were harvested, fixed in 10% neutral-buffered formalin, embedded in paraffin, and 3-µm sections were stained with hematoxylin and eosin. Tissue sections were examined by a pathologist blinded to experimental groups in a Leica DM2500 microscope coupled to a Leica MC170 HD microscope camera.

**Statistical analysis**. Comparisons of data between two groups were analyzed with the Student's *t* test (two-tailed, unpaired) or as detailed in the corresponding figure legend.

**Data availability**. The data that support the findings of this study are available from the corresponding author upon reasonable request.

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

## Acknowledgements

We thank members of the Carmo-Fonseca laboratory, in particular Sandra Martins and Teresa Carvalho for their insightful discussions and advice. We are also grateful to José Rino for help with the confocal microscope, and to Joana Marques and colleagues for animal husbandry. We further acknowledge Manuel Serrano, Sandrina Nóbrega, and Dafni Chondronasiou (CNIO, Spain) for their valuable comments and for providing the reprogrammable i4F mouse, i4F tissues, E14, and JCW9545 cell lines and Austin Smith (University of Edinburgh, UK) for sharing the TNG-A cell line. This work was supported by an AXA research fund, Fundação para a Ciência e Tecnologia (FCT), and FEDER (PTDC/BIM-MED/0032/2014; PTDC/BEX-BCM/5899/2014; and LISBOA-01-0145-FEDER-007391, a project cofunded by FEDER, through POR Lisboa 2020—Programa Operacional Regional de Lisboa, PORTUGAL 2020, and Fundação para a Ciência e a Tecnologia). B.B.d.J. is an FCT investigator (IF/00166/2014). C.A.-V. was a Gulbenkian Foundation Fellow. The funders had no role in study design, data collection and analysis, decision to publish, or preparation of the manuscript.

## Author contributions

B.B.d.J. and M.C.-F. conceived the study and wrote the paper. B.B.d.J. designed the experimental strategy and performed most of the experiments. S.P.M., A.S.-F., and C.A.-V. contributed to molecular biological analysis. S.B. contributed to cell culture analysis. T.C. performed the histopathological analysis of the subcutaneous tumors.

## Additional information

**Competing interests:** The authors declare no competing financial interests.

