## [Peer Review File · Nature Communications]

Reviewers' comments:

Reviewer #1 (Remarks to the Author):

Comments on 'Silencing the long non-coding RNA antisense to Zeb2 facilitates reprogramming of aged fibroblasts and safeguards stem cell pluripotency'

The authors found a mechanism for the inferior reprogramming of old fibroblasts versus young ones. They show that the overexpression of Zeb2 via its long non-coding RNA Zeb2-NAT inhibits reprogramming, and that silencing Zeb2-NAT facilitates reprogramming.

This is an interesting and important study with a clear and simple message. It has mechanistic as well as potentially practical implications. However, at its present form, there are several issues which require attention.

The major issue involves some confusing results, which the authors did not address. For example, the expression level of Zeb2-NAT in Fig. 3c shows a surprising pattern. In Fig. 1 the authors show that Zeb2-NAT is high in MEFs and low in iPSCs, but then, in Fig. 3c, Zeb2-NAT is low in 2i, then activated early in differentiation (20 hrs), and then low again after 72 hours. How does this fit with the high expression in MEFs, and how do the levels in MEFs and other fibroblasts compare with what is shown in Fig. 3c? This result also put the model in general in question. If Zeb2-NAT is low early in differentiation how can it correlate with pluripotency? This is confusing. Along the same lines, the authors show that 20 hours following LIF withdrawal, Zeb2-NAT is already high but Nanog remains unchanged, suggesting that upregulation of Zeb2-NAT precedes the down-regulation of Nanog/pluripotency gene expression. They also show that Nanog is already significantly reduced after 48 hours (Fig. 4) and 72 hours (Fig. 3), but that Zeb2-NAT is back to its low levels after 72 hours, somewhat confusing matters. The question is what happens after 48 hours to Zeb-NAT? The dynamics of immediate up-regulation followed by a significant down-regulation back to basal levels, all within the first 72 hours of spontaneous differentiation is unique, and since it goes both directions (Zeb2-NAT is low both in pluripotent and early differentiated cells) the relevance to pluripotency is not entirely clear.

Another confusing issue is the effect on teratoma formation. As a side-note, Fig. 4g&h are not at all cited in the text, and Fig.4h even has no figure legend. But the main problem is that it is unclear how the authors get no teratomas at all in the '-2i' (+LIF) conditions (??) ES cells should always give teratomas regardless of their growth conditions, especially in the presence of LIF (although even in the absence of LIF they do). This calls this whole experiment into question and the authors' expertise in ES cells. It is also not indicated how many teratomas were scored, whether results are significant, etc. It is also not clear how the Zeb2 treatment produced less teratomas than controls (??), especially given that in Fig. 4c, e - the authors show that the Zeb2 treatment was as effective as the Zeb2-NAT treatment in elevating levels of Nanog and E-cadherin.

Minor:

Fig. S1 should be quantified.

Fig. 1b. Indicate number of colonies from number of cells.

Fig. 1c. A single beta-catenin stained field does not provide any information. Beta-catenin staining should be compared between old/young and during the reprogramming process.

Fig. 1d, e, g. should be completed with data for 70 weeks.

Fig. 1e. IF control should be added (for a protein that does not change expression).

Fig. 1h. The authors can look for the expression of the studied transcripts in (several) available datasets and not settle with “an established iPSC cell line”. Also, the authors should add reprogrammed cells from the experiment, not a random iPSC colony.

In Fig. 2 the authors show that the fibroblasts treated with the anti-Zeb2-NAT oligonucleotides generated genuine iPSCs, while the control fibroblasts did not produce colonies at all. However, the reported efficiency of old fibroblasts in Fig. 1 (and in the literature) is not zero. The fact that they did not receive any colonies likely reflects the cell numbers used or other conditions (transfection?) used in the experiment. The authors should test the capacity of the rare iPSC colonies produced from the old fibroblasts to form teratomas. While obviously the reprogramming efficiency is compromised in the old cells, it is not clear whether, when successful, they can produce bona fide iPSCs.

In Fig. 3a the Zeb2-NAT is “barely detected” in E14 ESCs, but then in Supplementary Fig. 3a-b the authors show an order of magnitude less Zeb2-NAT when LNA gapmers are used, and a fair amount of detection.

Fig. 3e. There is no agreement between the text in the ms, which indicates a 24 hr time point and the text in the figure legend, which indicates a 48 hr time point. In any event, it is not clear why the authors used either. They should show here the 72 hr time point, to match the logic of the figure.

Fig. 4a,b – should also show Zeb2.

Reviewer #2 (Remarks to the Author):

In this manuscript Bernardes de Jesus and colleagues investigate the role on cell reprogramming of the regulators of the epithelial-to-mesenchymal transition Zeb2 and Zeb2-NAT, and how it relates with the age of the reprogrammed fibroblasts. They find that Zeb2 and Zeb2-NAT downregulation is stronger in younger fibroblasts when they are reprogrammed, and this downregulation favors reprogramming. They also conclude that Zeb2-NAT, which had been previously shown to regulate Zeb2, is an early marker of cell pluripotency.

This study addresses the interesting question of gene regulation during cellular reprogramming. However, the novelty of the study, which is the relationship between Zeb2 and its antisense noncoding transcript Zeb2-NAT, remains insufficiently explained. In addition, some of the conclusions are based on insufficient amount of data or lack controls.

Specific comments.

1. It is essential to determine how the inhibition of Zeb2-NAT is able to affect Zeb2 expression. In human cells it has been previously shown that Zeb2-NAT regulates Zeb2 splicing, resulting in Zeb2 increased translation due to the retention of an IRES. However, as far as I know, this mechanism hasn't been demonstrated as conserved in mouse cells. The authors claim that the increased expression of Zeb2-NAT in fibroblasts correlates with retention of the first intron of Zeb2 transcripts, however these data are not shown. In addition, the authors observe that the levels of Zeb2 mRNA are affected by Zeb2-NAT knockdown. In every experiment the authors should clearly show the relative levels of the different Zeb2 isoforms as well as the total mRNA and Zeb2 protein levels. Is Zeb2 splicing affected? Is Zeb2 regulated transcriptionally? Is it a combination of both?

2. Also related to the previous point, a problem of the study is the use of only one gapmer to inhibit Zeb2 and two for Zeb2-NAT. Furthermore, the gapmers that inhibit Zeb2-NAT are very close to Zeb2 TSS. It is possible that the effects observed on Zeb2 mRNA expression are due to

the interference of the gapmers with Zeb2 transcription (not just to Zeb2-NAT degradation). In addition, conclusions such as that the inhibition of Zeb2-NAT has more effect in pluripotency than the inhibition of Zeb2 could be affected by the different efficiencies of the gapmers knocking down the genes. More gapmers (or siRNAs) localized in other positions should be used to address the role of Zeb2-NAT and Zeb2. Similarly, rescue by ectopic expression of Zeb2 and Zeb2-NAT should be performed.

3. The authors have to be cautious when interpreting the reprogramming behavior and gene expression changes between MEFs and adult fibroblast as consequence of their age difference. MEFs and adult fibroblasts are not strictly the same cell type only differing in age. Reprogramming with adult fibroblast from adults of different ages should be shown in more experiments (for instance fig 1D) to proof this point.

4. The characterization of the phenotype of reprogrammed cells should be consistent across all experiments. Reprogramming efficiency in some experiments is quantified based on AP staining, while in others additional filters are used.

5. It is not clear how in several experiments the authors are able to do statistical analysis with only two replicates.

Reviewer #1:

The major issue involves some confusing results, which the authors did not address. For example, the expression level of Zeb2-NAT in Fig. 3c shows a surprising pattern. In Fig. 1 the authors show that Zeb2-NAT is high in MEFs and low in iPSCs, but then, in Fig. 3c, Zeb2-NAT is low in 2i, then activated early in differentiation (20 hrs), and then low again after 72 hours. How does this fit with the high expression in MEFs, and how do the levels in MEFs and other fibroblasts compare with what is shown in Fig. 3c? This result also put the model in general in question. If Zeb2-NAT is low early in differentiation how can it correlate with pluripotency? This is confusing.

We thank the reviewer for alerting to this apparent confusion. We carried out more experiments after 72 h in culture in medium devoid of LIF and 2i, and the results show that levels of Zeb2-NAT are not statistically different at 20 and 72 h (new Fig. 6c). We also rewrote the text and added more experiments to show that Zeb2-NAT is low when stem cells are in the ground state of pluripotency but its expression is activated as soon as stem cells are challenged to differentiate (new Fig. 6, Fig. S7). We also now show the relative levels of Zeb2-NAT in fibroblasts and stem cells before and after being challenged to differentiate (new Fig. 6 d).

Along the same lines, the authors show that 20 hours following LIF withdrawal, Zeb2-NAT is already high but Nanog remains unchanged, suggesting that upregulation of Zeb2-NAT precedes the down-regulation of Nanog/pluripotency gene expression. They also show that Nanog is already significantly reduced after 48 hours (Fig. 4) and 72 hours (Fig. 3), but that Zeb2-NAT is back to its low levels after 72 hours, somewhat confusing matters. The question is what happens after 48 hours to Zeb-NAT? The dynamics of immediate up-regulation followed by a significant down-regulation back to basal levels, all within the first 72 hours of spontaneous differentiation is unique, and since it goes both directions (Zeb2-NAT is low both in pluripotent and early differentiated cells) the relevance to pluripotency is not entirely clear.

We did more experiments addressing the dynamics of immediate up-regulation of Zeb2-NAT at 20h and 48h (new Fig. 6 d). The results show a steady increase at both time points.

Another confusing issue is the effect on teratoma formation. As a side-note, Fig. 4g&h are not at all cited in the text, and Fig.4h even has no figure legend. But the main problem is that it is unclear how the authors get no teratomas at all in the '-2i' (+LIF) conditions (??) ES cells should always give teratomas regardless of their growth conditions, especially in the presence of LIF (although even in the absence of LIF they do). This calls this whole experiment into question and the authors' expertise in ES cells. It is also not indicated how many teratomas were scored, whether results are significant, etc.

The reviewer is absolutely correct. Teratomas do form upon injection of E14 cells grown either in LIF/2i or LIF only (see new Fig. S8). In contrast, the transgenic line TNG-A is sensitive to removal of 2i, as previously reported (Descalzo et al. Stem Cells 2012;30:2683–2691). As suggested by the reviewer, we did more teratoma experiments and the results are now presented in detail in new Fig. S10.

It is also not clear how the Zeb2 treatment produced less teratomas than controls (??), especially given that in Fig. 4c, e – the authors show that the Zeb2 treatment was as effective as the Zeb2-NAT treatment in elevating levels of Nanog and E-cadherin.

We clarify that teratomas were detected in animals injected with cells treated with either anti-Zeb2-NAT or anti-Zeb2 oligonucleotides. However, tumors were more prominent and more differentiated when Zeb2-NAT was targeted (new Fig. 7g and Fig. S10), probably because LNA Gapmers were more efficient in knocking down Zeb2-NAT transcripts than Zeb2 mRNA.

Fig. S1 should be quantified.

The quantification of AP+ colonies and reprogramming efficiency is now presented in new Fig. 1b, c, Fig. 4 and Fig. S4.

Fig. 1b. Indicate number of colonies from number of cells.

This is now provided (new Fig. 1b).

Fig. 1c. A single beta-catenin stained field does not provide any information. Beta-catenin staining should be compared between old/young and during the reprogramming process.

We thank the reviewer for this suggestion. This is now presented in Fig. S1b.

Fig. 1d, e, g. should be completed with data for 70 weeks.

Data for fibroblasts from 70-100 week-old mice is now included in new Fig. 1d, e, g.

Fig. 1e. IF control should be added (for a protein that does not change expression).

We thank the reviewer for this suggestion. This is now presented in Fig. S1c.

Fig. 1h. The authors can look for the expression of the studied transcripts in (several) available datasets and not settle with “an established iPSC cell line”. Also, the authors should add reprogrammed cells from the experiment, not a random iPSC colony.

As suggested by the reviewer, we now show data for iPSCs obtained in our own experiments (new Fig. 2c).

In Fig. 2 the authors show that the fibroblasts treated with the anti-Zeb2-NAT oligonucleotides generated genuine iPSCs, while the control fibroblasts did not produce colonies at all. However, the reported efficiency of old fibroblasts in Fig. 1 (and in the literature) is not zero. The fact that they did not receive any colonies likely reflects the cell numbers used or other conditions (transfection?) used in the experiment. The authors should test the capacity of the rare iPSC colonies produced from the old fibroblasts to form teratomas. While obviously the reprogramming efficiency is compromised in the old cells, it is not clear whether, when successful, they can produce bona fide iPSCs.

The reviewer is correct. We did more reprogramming experiments and it was indeed possible to obtain iPSC colonies from old fibroblast (new Fig. 1b, c and Fig. S4). We also

clarify in the revised text that iPS-like colonies derived from old fibroblasts did not grow in feeder-free culture, but could be expanded through the use of feeders (new Fig. S4d).

In Fig.3a the Zeb2-NAT is “barely detected” in E14 ESCs, but then in Supplementary Fig. 3a-b the authors show an order of magnitude less Zeb2-NAT when LNA gapmers are used, and a fair amount of detection.

We thank the reviewer for this alert. The text was revised accordingly. See also new Fig. 6 and Fig. S9.

Fig. 3e. There is no agreement between the text in the ms, which indicates a 24 hr time point and the text in the figure legend, which indicates a 48 hr time point. In any event, it is not clear why the authors used either. They should show here the 72 hr time point, to match the logic of the figure.

This has been corrected in the revised manuscript.

Fig. 4a,b – should also show Zeb2.

Data on Zeb2 has been included in new Fig. S9d.

Reviewer #2:

It is essential to determine how the inhibition of Zeb2-NAT is able to affect Zeb2 expression. In human cells it has been previously shown that Zeb2-NAT regulates Zeb2 splicing, resulting in Zeb2 increased translation due to the retention of an IRES. However, as far as I know, this mechanism hasn't been demonstrated as conserved in mouse cells. The authors claim that the increased expression of Zeb2-NAT in fibroblasts correlates with retention of the first intron of Zeb2 transcripts, however these data are not shown. In addition, the authors observe that the levels of Zeb2 mRNA are affected by Zeb2-NAT knockdown. In every experiment the authors should clearly show the relative levels of the different Zeb2 isoforms as well as the total mRNA and Zeb2 protein levels. Is Zeb2 splicing affected? Is Zeb2 regulated transcriptionally? Is it a combination of both?

We thank the reviewer for raising these important questions. In the revised manuscript we now include more experiments showing that increased expression of Zeb2-NAT correlates with retention of the first intron of Zeb2 transcripts. In particular, we now show the relative levels of the different Zeb2 spliced and unspliced isoforms (new Fig. 2d and 3d), as well as the total Zeb2 protein levels (new Fig. 1g, 3c, 5d). We also used digital PCR to obtain a more precise absolute quantification of the different types of RNA molecules (new Fig. S3a). In addition, we did metabolic labelling experiments with 4SU to specifically analyse newly synthesized RNA (new Fig. S3b). The results indicate that Zeb2 is regulated at the level of both splicing and transcription.

Also related to the previous point, a problem of the study is the use of only one gapmer to inhibit Zeb2 and two for Zeb2-NAT. Furthermore, the gapmers that inhibit Zeb2-NAT are very close to Zeb2 TSS. It is possible that the effects observed on Zeb2 mRNA expression are due to the interference of the gapmers with Zeb2 transcription (not just to Zeb2-NAT degradation). In addition, conclusions such as that the inhibition of Zeb2-NAT has more effect in pluripotency than the inhibition of Zeb2 could be affected by the different efficiencies of the gapmers knocking down the genes. More gapmers (or siRNAs) localized in other positions should be used to address the role of Zeb2-NAT and Zeb2. Similarly, rescue by ectopic expression of Zeb2 and Zeb2-NAT should be performed.

As suggested by the reviewer, we used shRNAs to knockdown Zeb2 and Zeb2-NAT by RNAi (new Fig. S5). We also now show rescue of the Gapmer-induced phenotype by ectopic expression of 2'OMe RNA oligonucleotides that mimic Zeb2-NAT sequences (new Fig. 5).

The authors have to be cautious when interpreting the reprogramming behavior and gene expression changes between MEFs and adult fibroblast as consequence of their age difference. MEFs and adult fibroblasts are not strictly the same cell type only differing in age. Reprogramming with adult fibroblast from adults of different ages should be shown in more experiments (for instance fig 1D) to proof this point.

Reprogramming with fibroblasts from adult animals of different ages (10 to 30 weeks old versus 70 to 100 weeks old) is now shown in new Fig. 1 and 2b.

The characterization of the phenotype of reprogrammed cells should be consistent across all experiments. Reprogramming efficiency in some experiments is quantified based on AP staining, while in others additional filters are used.

This has been corrected throughout the manuscript.

It is not clear how in several experiments the authors are able to do statistical analysis with only two replicates.

We repeated many experiments. Panels throughout the manuscript now depict data from at least three independent biological replicates.

REVIEWERS' COMMENTS:

Reviewer #1 (Remarks to the Author):

The authors addressed the concerns and the manuscript is now significantly improved. The only minor remaining issue is the quantification of Supplementary Figure S1 which the authors should add. Otherwise this interesting manuscript is acceptable for publication.

Reviewer #2 (Remarks to the Author):

Overall the authors have done a good job. Although they haven't shown the rescue by expressing Zeb2 and Zeb2-NAT, I consider that the data presented are robust enough to support their conclusions.

Reviewer #1:

The authors addressed the concerns and the manuscript is now significantly improved. The only minor remaining issue is the quantification of Supplementary Figure S1 which the authors should add. Otherwise this interesting manuscript is acceptable for publication.

We thank the reviewer for the commentaries. In the final revised version of our manuscript we add the quantification of Supplementary Figure S1 (referred as Supplementary Figure S1b)

Reviewer #2:

Overall the authors have done a good job. Although they haven't shown the rescue by expressing Zeb2 and Zeb2-NAT, I consider that the data presented are robust enough to support their conclusions.

We thank the reviewer for the positive commentaries.